# Enrichment of rare codons at 5' ends of genes is a spandrel caused by evolutionary sequence turnover and does not improve translation

Richard Sejour[1], Janet Leatherwood[2], Alisa Yurovsky[3], Bruce Futcher[2]*

[1]Department of Pharmacological Sciences, Stony Brook University, Stony Brook, United States; [2]Department of Microbiology and Immunology, Stony Brook University, Stony Brook, United States; [3]Department of Biomedical Informatics, Stony Brook University, Stony Brook, United States

*For correspondence:
bfutcher@gmail.com

Competing interest: The authors declare that no competing interests exist.

**Abstract** Previously, Tuller et al. found that the first 30–50 codons of the genes of yeast and other eukaryotes are slightly enriched for rare codons. They argued that this slowed translation, and was adaptive because it queued ribosomes to prevent collisions. Today, the translational speeds of different codons are known, and indeed rare codons are translated slowly. We re-examined this 5' slow translation 'ramp.' We confirm that 5' regions are slightly enriched for rare codons; in addition, they are depleted for downstream Start codons (which are fast), with both effects contributing to slow 5' translation. However, we also find that the 5' (and 3') ends of yeast genes are poorly conserved in evolution, suggesting that they are unstable and turnover relatively rapidly. When a new 5' end forms de novo, it is likely to include codons that would otherwise be rare. Because evolution has had a relatively short time to select against these codons, 5' ends are typically slightly enriched for rare, slow codons. Opposite to the expectation of Tuller et al., we show by direct experiment that genes with slowly translated codons at the 5' end are expressed relatively poorly, and that substituting faster synonymous codons improves expression. Direct experiment shows that slow codons do not prevent downstream ribosome collisions. Further informatic studies suggest that for natural genes, slow 5' ends are correlated with poor gene expression, opposite to the expectation of Tuller et al. Thus, we conclude that slow 5' translation is a 'spandrel'--a non-adaptive consequence of something else, in this case, the turnover of 5' ends in evolution, and it does not improve translation.

## eLife assessment

This is an **important** contribution to the origins and translational consequences of the relatively low rate of translation elongation in the first ~30-50 codons of genes in most organisms. The authors provide **convincing** evidence that the prevalence of rare codons in the first ~40 codons in yeast is due to the relatively recent evolution of these coding sequences, or of lower purifying selection operating on them, and that a preponderance of codons encoded by rare tRNAs near the N-terminus is not associated with higher translational efficiency in the manner proposed by the "translational ramp" hypothesis. The work is **incomplete** in that the results of reporter assays may have been confounded by alterations of mRNA sequence or structure that could have influenced their translation or mRNA stability; that the work cannot fully account for a greater enrichment of slowly translated codons in N-terminal vs. C-terminal regions; and that the work does not resolve whether translation elongation through N-terminal coding is truly slow.

## Introduction

(*Tuller et al., 2010*) were interested in the idea that a slow translational ramp at the beginning of a gene might queue ribosomes in an orderly way, thereby preventing ribosome traffic jams and collisions. However, at that time, translation speeds for the 61 sense codons were not known from direct measurement. Therefore, as a proxy for codon translation speed, Tuller et al. devised a proxy speed measurement based on the tRNA-adaptation index (tAI), a measure of the abundance of each tRNA. The assumption is that codons recognized by more abundant tRNAs would be translated faster. Using this proxy, Tuller et al. found that in yeast and other eukaryotes, the first 30–100 codons are enriched for codons for which tRNAs are rare (generally, rare codons), and are presumably translated slowly. The size of the effect is small (about a 3% difference, Figure 2C of *Tuller et al., 2010*), but is statistically highly significant.

At the time of the work of Tuller et al. ribosome profiling had recently been developed (*Ingolia et al., 2009*), and early ribosome profiling showed a high density of ribosomes near the 5' end of the mRNA, consistent with slow translation in this region, and the rare codons found by Tuller et al. could have contributed to this. However, later work showed that this 5' high density of ribosomes was an artifact of the way cycloheximide was used to arrest translation in the original protocol (*Weinberg et al., 2016*). With newer protocols for ribosome profiling, which use cycloheximide only at later steps, the region of 5' high ribosome density largely, but not entirely, disappears (*Weinberg et al., 2016*) (see Discussion).

Since then, many workers have used ribosome profiling to directly measure the speed of translation of individual codons (cited below). With such data in hand, we revisited the issues addressed by *Tuller et al., 2010*. On the one hand, our analyses confirm that the 5' regions of genes are typically slightly enriched for rare codons, and these encodings likely slow translation. On the other hand, various aspects of the data led us to an alternative hypothesis, namely that the 5' ends were turning over relatively rapidly in evolution; that these 5' ends were, therefore, relatively young; and that selection had not yet succeeded in removing all the rare, slow codons initially present in the de novo 5' ends. We did a direct experimental test of the effects of slow or fast initial translation. Opposite to Tuller et al., we found that encoding slow initial translation resulted in lower protein production than fast initial translation. This continued to be true even when we placed ribosome collision sites inside the reporter gene. Thus a slow initial translation ramp, though present, neither improves gene expression nor prevents ribosome collisions.

It is natural to assume that the enrichment of slow codons near 5' ends is a product of selection. However, as elegantly argued by *Gould and Lewontin, 1979* in their classic paper 'The Spandrels of San Marco and the Panglossian Paradigm: A Critique of the Adaptionist Programme,' not all biological phenomena are adaptive, or even a direct product of selection. They argued from the example of a 'spandrel:' in architecture, a triangular space created when an arch supports a lintel. There is no architectural role for spandrels as such; they are the indirect and inevitable result of the juxtaposition of two other functional architectural elements. We argue that the slightly slow initial translation of eukaryotic genes may likewise be a spandrel, a non-adaptive consequence of something else, the instability of 5' ends in evolution.

## Results

### Calculations of encoded translation speed imply slow initial translation

*Tuller et al., 2010* used codon-specific tRNA abundance as a proxy to estimate the speed of translation of codons. Since then, analysis of ribosome profiling data has yielded direct measurements of the translation speed of individual codons (*Weinberg et al., 2016*; *Dao Duc and Song, 2018*; *Gardin et al., 2014*; *Gritsenko et al., 2015*; *Lareau et al., 2014*; *Sharma et al., 2019*; *Tunney et al., 2018*; *Wang et al., 2017*). Accordingly, we have repeated some of the work of Tuller et al. but using the Ribosome Residence Time (RRT, Methods and materials, *Supplementary file 1*; *Gardin et al., 2014*) of each of the 61 sense codons as a measure of translation speed. We refer to 'encoded translation speed' to specify that we are focusing purely on the effects of different codons on translation speed, and not on other factors that might differentially affect translation speed at different regions of the mRNA, such as secondary structure.

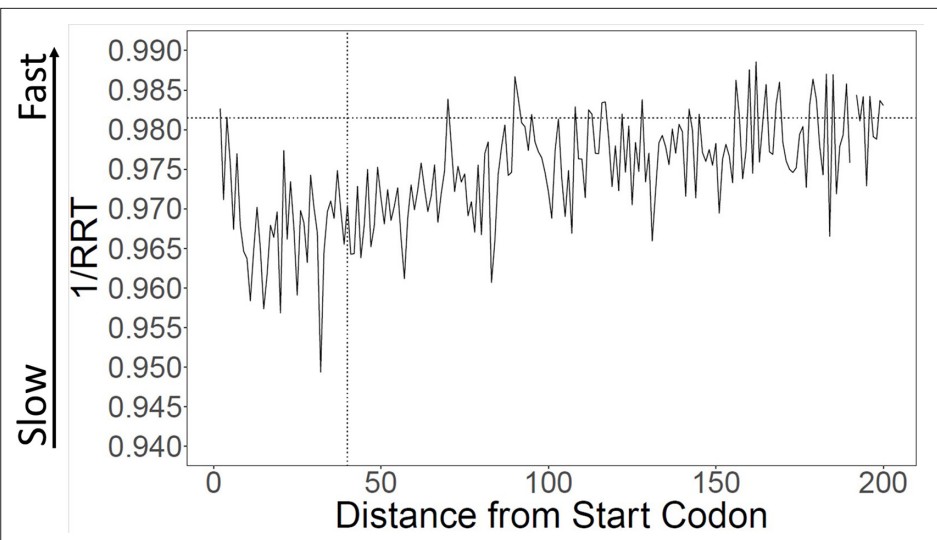

**Figure 1.** Calculation of translation speed confirms slow initial translation (SIT). Translation speeds were calculated using ribosome residence time (RRT) (*Gardin et al., 2014*; *Supplementary file 1* for RRT values) as a measure of codon-specific translation speed over *S. cerevisiae* open reading frames (ORFs). The horizontal line indicates average inverse RRT across all ORFs. The average speed in the first 40 amino acids is about 1.1% slower than in the rest of the gene (p < 0.001).

The online version of this article includes the following figure supplement(s) for figure 1:

**Figure supplement 1.** Distribution of translation speeds at 5' and 3' ends.

Using the RRT, encoded translation speeds were calculated in sliding windows across all coding ORFs from *S. cerevisiae*. The start codon was omitted because it is constant across genes and is an unusually 'fast' codon. Consistent with Tuller et al. we find that the first 30–100 codons had lower calculated translation speeds than the rest of the gene (*Figure 1*). We focused our analyses on the first 40 codons for comparability to Tuller et al.; we call this the 'Slow Initial Translation' region, or SIT. Although the tendency towards slow translation near the beginning of the gene is very highly statistically significant, the size of the effect is small. When we compare the first 40 codons of a gene to the rest of the same gene, we find that the average difference in encoded translation speed is about 1.2% (similar to *Tuller et al., 2010*), with a p-value <0.001. For comparison, codons can vary in translation speed by about threefold (*Supplementary file 1*), or perhaps as much as sixfold (*Weinberg et al., 2016*, their Table S2).

Although on average genes are translated slowly near their 5' ends, there is variability. *Figure 1—figure supplement 1* shows the distribution of initial encoded translation speed for all yeast ORFs. About 57% of genes have a slow initial translation (SIT) region, while the remainder have fast initial translation (FIT).

## Rare (slow) codons are enriched within the first 40 codons

Rare codons tend to be slow codons, and *vice versa* (*Gardin et al., 2014*; *Figure 2—figure supplement 1*). But the correlation is not perfect—other things being equal, A/T-rich codons tend to be faster than G/C-rich codons, and codons with a third position wobble base tend to be faster than codons with the cognate base (*Gardin et al., 2014*). To better understand why gene beginnings are more slowly translated, we examined the relative usage of each of the 61 sense codons in the first 40 codons after but not including the initiator ATG (*Figure 2*).

For almost all amino acids, with isoleucine (ATC, ATA, ATT) being the only clear exception, we saw relative enrichment of the rarest and generally slowest codons. In particular, there were notable enrichments of the three rarest, slowest arginine codons (CGA, CGC, CGG) (likely because of their use in N-terminal signal sequences, see below) and the slow, rare codons for proline (CCC, CCG), leucine (CTC), glycine (GGG), and cysteine (TGC) (*Figure 2A*). These enrichments can explain most of the slow initial translation.

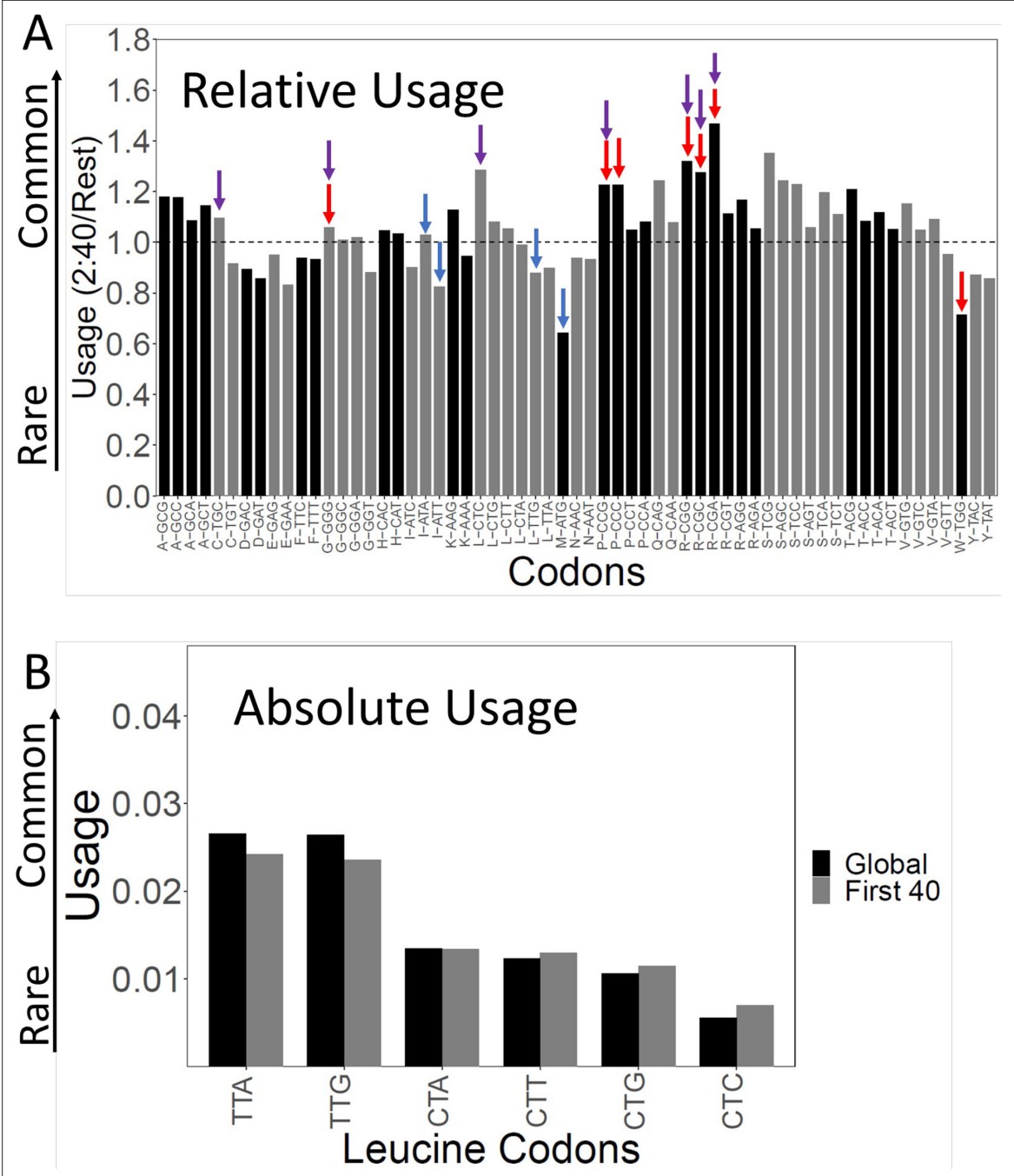

**Figure 2.** Codon usage in the slow initial translation (SIT) region. (**A**). Relative codon usage in the SIT versus the rest of the gene. The Y-axis shows codon usage in the first 40 amino acids (omitting ATG) divided by its usage in the rest of the gene. The 61 sense codons are grouped by amino acid. Within each group, codons are ordered from least to most frequent left to right. Red arrows show the seven slowest codons by ribosome residence time (RRT), purple arrows show the seven rarest codons by total usage, and *Figure 2—figure supplement 1* shows the correlation between codon usage and translation speed. Blue shows Start and alternative Start codons (ATG, TTG, ATT, ATA). Ratios above 1 show enrichment in the first 40 amino acids. Typically, the rarest codons are enriched. (**B**). Absolute usage of each leucine codon in the SIT. The absolute usage frequency of each leucine codon is shown globally, and for the first 40 amino acids. Rare codons are still rare in the SIT, just not as rare as elsewhere. The same pattern holds for the other amino acids.

The online version of this article includes the following figure supplement(s) for figure 2:

**Figure supplement 1.** Codon speed and codon usage are correlated.

In addition, in the first 40 codons after but not including the initiator ATG, we saw notable depletion of the canonical Start codon ATG (by nearly 50%), and the alternative Start codons ATT and TTG (*Eisenberg et al., 2020*; *Figure 2A*) by lesser but still significant amounts. These three codons are very fast (*Supplementary file 1*), and the depletion of these fast codons would make the average translation speed slower. Possibly Start codons are depleted to reduce the possibility of translation initiating at the wrong place. We recalculated translation speeds after assigning ATG a neutral speed (*Supplementary file 2*). This reduced the difference in translation speed between the SIT and the rest of the gene by about 15% of the difference. Since the three most commonly used alternative Start codons are ATT, TTG, and ATA (*Eisenberg et al., 2020*), we also neutralized these by assigning them a neutral RRT (1.0189), in addition to neutralizing ATG. After neutralization of all four codons, the difference in translation speed between the SIT and the rest of the gene was reduced by about 40%, a significant change (*Supplementary file 2*). Even so, the remaining slow initial translation was highly significant. Thus, the depletion of Start codons contributes significantly to slow initial translation, but enrichment for rare, slow codons contributes even more.

(ATG and alternative Start codons are also depleted in the other two reading frames, but these depletions have indirect effects on the in-frame codons, such that there are roughly off-setting effects on translation speed. For instance, the depletion of TGG (Trp) (*Figure 2*) is likely partly due to depletion of xx**A TG**G, but since TGG is a slow codon, this depletion increases 5′ translation speed.)

## Rare codons are rare in the first 40 codons, just not as rare as elsewhere

Although the proportion of rare, slow codons in the SITs is relatively higher than in the body of genes, in absolute terms rare codons are still rare compared to more common synonymous codons (e.g. *Figure 2B*, Leu codons). That is, rare, slow codons are still strongly disfavored in the SIT, though they are less strongly disfavored than elsewhere. This was true for all rare codons.

## Why are there relatively more rare, slow codons at 5′ ends?

Our analysis is consistent with that of Tuller et al. to the extent that we find a slight relative enrichment of rare, slow codons near the 5′ ends of coding regions. Tuller et al. interpret the slow translation ramp as an adaptation—they believe there is a selection for slow codons near the 5′ end to enhance the efficiency of translation. But there are other possibilities.

## The Young Spandrel hypothesis

We noticed (see below) that the N-termini of yeast genes are often poorly conserved, and otherwise highly homologous genes often vary at the N-termini between different closely related species. This suggests a different idea: N-termini are unstable and variable in evolution. They form de novo from new DNA sequence, and so all codons may initially occur at similar frequencies. De novo formation of a N-terminus could occur from use of a new Start codon (*Bazykin and Kochetov, 2011*; *Kochetov, 2008*). Since these N-termini are, on average, younger than the remainder of the gene, selection has worked on them for a shorter time. Therefore, selection against rare, slow codons may be less complete, and N-termini, due to their relative youth, may still retain some extra rare codons.

In this idea, in contrast to Tuller et al. the slight excess of rare codons near N-termini is not adaptive; it is not at all a product of selection. Instead, in the words of *Gould and Lewontin, 1979*, it is a spandrel. It is a non-adaptive by-product of something else, in this case, the evolutionary instability of N-termini. This idea is explored below.

## Poor 5′ conservation is a feature of many yeast genes

We picked example genes for illustration. We used protein-protein BLAST at NCBI to blast several query genes against species of the subphylum *Saccharomycotina* (but having subtracted out all of *Saccharomyces*). *Figure 3* shows that for these examples, the middle portions of the proteins are highly conserved, but the N-termini are not. We suggest that these non-conserved N-terminal regions are young in evolution, and therefore likely contain an excess of rare codons.

In the next section, we ask whether the findings of *Figure 3* can be generalized to proteins of *S. cerevisiae*.

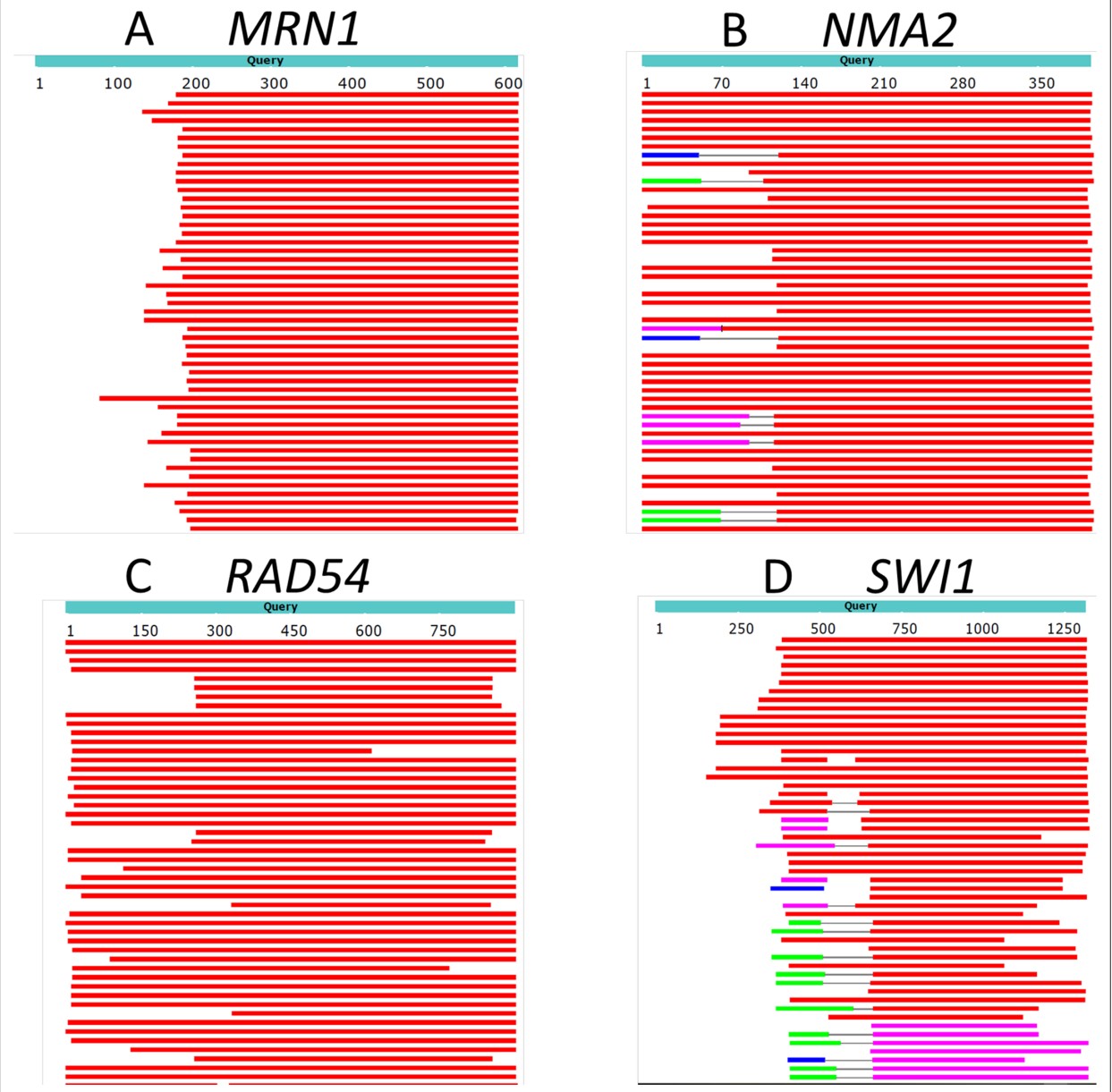

**Figure 3.** The N-termini of proteins can vary in evolution. BLAST of four example *S. cerevisiae* proteins against proteins in the subphylum '*Saccharomycotina*' (taxid: 147537) (excluding *Saccharomyces,* taxid 4930) was performed. Top hits are shown. Red regions indicate homology with an alignment score >200, while white indicates no detected homology (BLAST default parameters). Even though all hits have high to moderate homology towards the center of the protein, many have little or no homology at the N-terminus.

## Method of scoring N-terminal conservation, and rationale for using *Saccharomycotina*

We investigated N-terminal protein conservation using a quantitative approach. We ran local protein-protein BLAST for all *S. cerevisiae* genes against sequences of the *Saccharomycotina* subphylum, omitting *Saccharomyces cerevisiae. Saccharomycotina* was chosen because almost every gene from *S. cerevisiae* has a recognizable, conserved homolog in almost every species in *Saccharomycotina*, and yet the evolutionary distances are long enough that there is considerable sequence variability. We excluded species of *Saccharomyces* as they are too closely related to *S. cerevisiae*, and they are very numerous in sequence collections, and would overwhelm results from the other members of *Saccharomycotina*. However, we believe that this choice of subphylum does not greatly affect the final result.

Conservation at the N-terminus was calculated as the weighted proportion of yeast species with sequence matches (a match by default BLAST parameters) beginning in the first 40 amino acids. The lowest conservation score is 0 (no hits in the first 40 amino acids), whereas the highest conservation score is 40 indicating that every species had a match (default BLAST parameters) starting at the first amino acid (*Supplementary file 3 and 4*). The length of genes is negatively correlated with the conservation score, especially at the N-terminus (rho = −0.47; p <0.001), but also for the rest of the gene (rho = −0.37, p<0.001)—that is, short genes tend to be more conserved.

## N-termini are variable and poorly conserved

We measured protein conservation across more than 3000 *S. cerevisiae* proteins with orthologues among 822 closely related yeasts from *Saccharomycotina*. For each protein, we developed a conservation score for the first 40 amino acids to represent the N-termini, an equivalent conservation score for the middle 40 amino acids, and an equivalent conservation score for the C-terminal 40 amino acids (Methods and materials; *Supplementary file 3 and 4*).

Strikingly, the N-termini of *S. cerevisiae* orthologs had conservation scores that were much lower, and very differently distributed than the middle of the same orthologs (*Figure 4*). The first 40 amino acids had a flat distribution of protein conservation scores, indicating high levels of variability amongst these orthologs. That is, many of the orthologs had no detectable homology with the first 40 amino acids of the *cerevisiae* protein.

In contrast, the middle 40 amino acids were highly conserved, with conservations scores peaking at 40, the highest possible score. It Is evident that for the middle 40 amino acids, a large fraction of orthologs had a region of high homology to the *S. cerevisiae* protein, whereas this was not true for the N-termini. Finally, the last 40 amino acids had conservation scores similar to those of the first 40 amino acids, though a bit higher (more conserved) see *Figure 4—figure supplement 1* for a comparison. These results suggest both ends of the gene 'breathe,' gaining and losing new sequences during evolution, whilst the middles stay constant. Thus the ends of genes are younger than their middles. At their first formation, they would likely have contained some rare codons, which selection may not yet have had time to remove.

## The 3' ends of genes also have slightly slow translation

As shown in *Figure 4C*, the C-termini of proteins have poor conservation, like the N-termini. Therefore, the Spandrel hypothesis predicts slow translation at 3' ends. We calculated translation speeds at 3' ends, and again found slightly slow translation (*Figure 5*). This was not statistically significant over the last 40 codons, but it was significant over the last 100 codons (*Figure 5*) and the last 120 codons (*Supplementary file 2*). Like the 5' end, there was a slightly increased relative frequency of rare codons (*Figure 5—figure supplement 1*), but unlike the 5' end, ATG was not depleted (*Figure 5—figure supplement 1*).

There may be at least three reasons why translation at 3' ends is not as slow as at 5' ends. First, at 3' ends, Start codons and alternative-Start codons (which are fast) are not depleted (because at 3' ends, there is no issue of generating incorrect translation initiation sites), and this retention of fast codons tends to make 3' ends faster than 5' ends.

Second, there are two sets of genes, mitochondrial genes, and ER (Endoplasmic Reticulum) genes, that have especially slow 5' translation. Tuller et al. characterized several functional groups of genes for the amplitude and length of their slow initial translation ramp. The group with the greatest amplitude was the group of 282 genes for 'Mitochondrial organization.' This group is dominated by genes for proteins imported into mitochondria. We believe this especially slow translation is due to N-terminal signal sequences. Mitochondrial import depends on an N-terminal signal sequence. A typical mitochondrial signal sequence has an average of 25 residues, is highly enriched for arginine, and has relatively little sequence conservation (e.g. cluster I of *Fukasawa et al., 2015*). Since four of the 10 slowest codons are for arginine, and since little sequence conservation is required in a mitochondrial signal sequence, these signal sequences seem good candidates for regions that could vary rapidly in evolution, and have slow initial translation (thanks largely to rare, slow Arg codons). Indeed, we found that for a set of 467 mitochondrial proteins (*Williams et al., 2014*) initial translation was about 2.3% slower than in the rest of the genes, versus only 1.04% slower for genes with no mitochondrial or ER signal sequence (*Supplementary file 2* and *Figure 5—figure supplement 2*). We had similar findings,

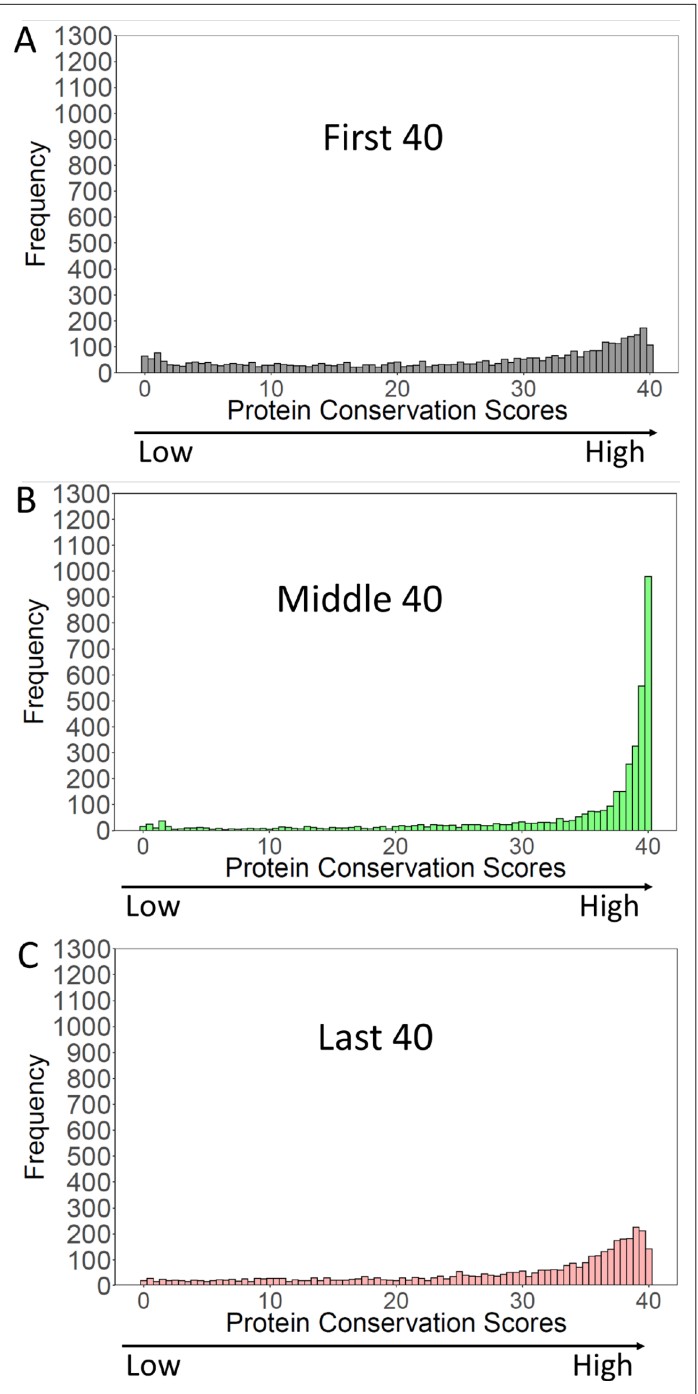

**Figure 4.** Conservation of *S. cerevisiae* proteins over the N-terminal, Middle, and C-terminal 40 amino acids. *S. cerevisiae* proteins were blasted against proteins of *Saccharomycotina* (excluding *cerevisiae*). 'Conservation Scores' (Methods and materials) were calculated for the N-terminal, Middle, and C-terminal 40 amino acids of the *S. cerevisiae* proteins. Scores range from 0 (no conservation) to 40 (perfect conservation). The frequency of each conservation score (3964 *S. cerevisiae* proteins) was plotted.

The online version of this article includes the following figure supplement(s) for figure 4:

**Figure supplement 1.** Comparison of conservation scores at the N- and C-termini.

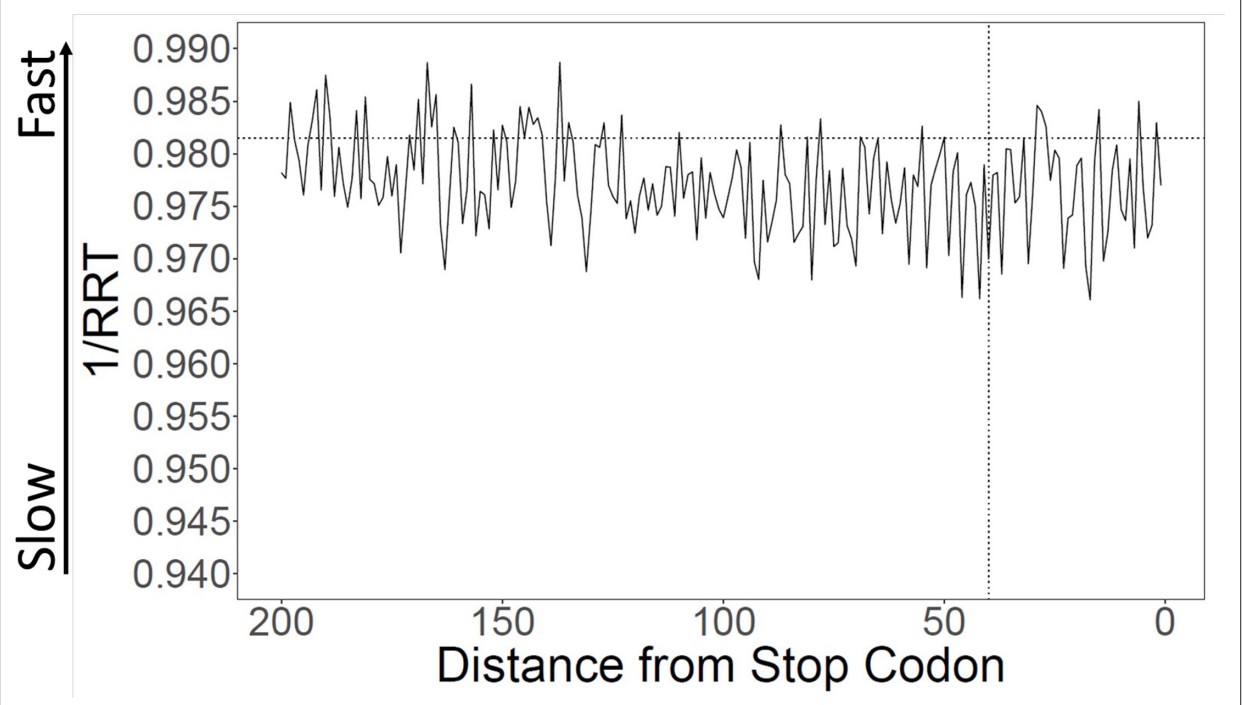

**Figure 5.** Translation speed at 3' ends. Translation speeds at the 3' ends of genes were calculated using ribosome residence time (RRT) (***Gardin et al., 2014***; ***Supplementary file 1*** for RRT values). The average speed over the last 40 amino acids is about 0.1% slower than in the rest of the gene, not statistically significant. The average speed over the last 100 amino acids is about 0.19% slower, which is significantly different (p=0.028).

The online version of this article includes the following figure supplement(s) for figure 5:

**Figure supplement 1.** Codon characteristics at the beginnings and ends of yeast genes.

**Figure supplement 2.** Mitochondrial and ER signal sequences.

but to a lesser extent, for proteins with an ER signal sequence, which is also rich in basic residues (***Figure 5—figure supplement 2***). In this case, ***Pechmann et al., 2014*** have argued that a cluster of rare, slow codons 35–40 codons from the N-terminus provide a translational pause that allows the Signal Recognition Particle time to recognize the signal sequence. Both kinds of explanations could be true.

Third, as an adaptation argument, 5' ends could sometimes be selected for poor translation to produce an appropriately small amount of protein, and this would sometimes favor rare codons (see below).

(***Cope et al., 2018***) had similar findings for N-terminal signal peptides of *E. coli*, which are enriched in translationally inefficient codons. Like us, they suggested selection for codon usage was relatively weak (or evolutionarily brief) at 5' ends, and they cited the 'Spandrel' idea of ***Gould and Lewontin, 1979***: that is, the inefficient codons might have arisen for a non-adaptive reason, and persisted because of weak (or brief) selection.

## 5' translation speeds positively correlate with 5' conservation scores

If the 'Young Spandrel' hypothesis is true, and slow 5' translation is partly caused by evolutionary instability of 5' ends, then there should be a correlation between encoded slow translation, and poor N-terminal conservation. Our model predicts the least conserved N-termini to have the slowest translation (i.e. rarest codons), and, *vice versa*, the termini with the slowest translation should have the lowest conservation. To test this, we ranked all genes by the conservation scores of their first 40 amino acids. We then divided this ranked list into thirds. For each of the thirds (i.e. the bottom, middle, and top conservation scores) we plotted the average relative initial translation speeds.

As shown in ***Figure 6A***, the genes with the most poorly conserved N-termini also had the slowest initial translation, while the genes with the most conserved N-termini had the fastest initial translation, supporting the Spandrel hypothesis, and opposite to the Ramp hypothesis.

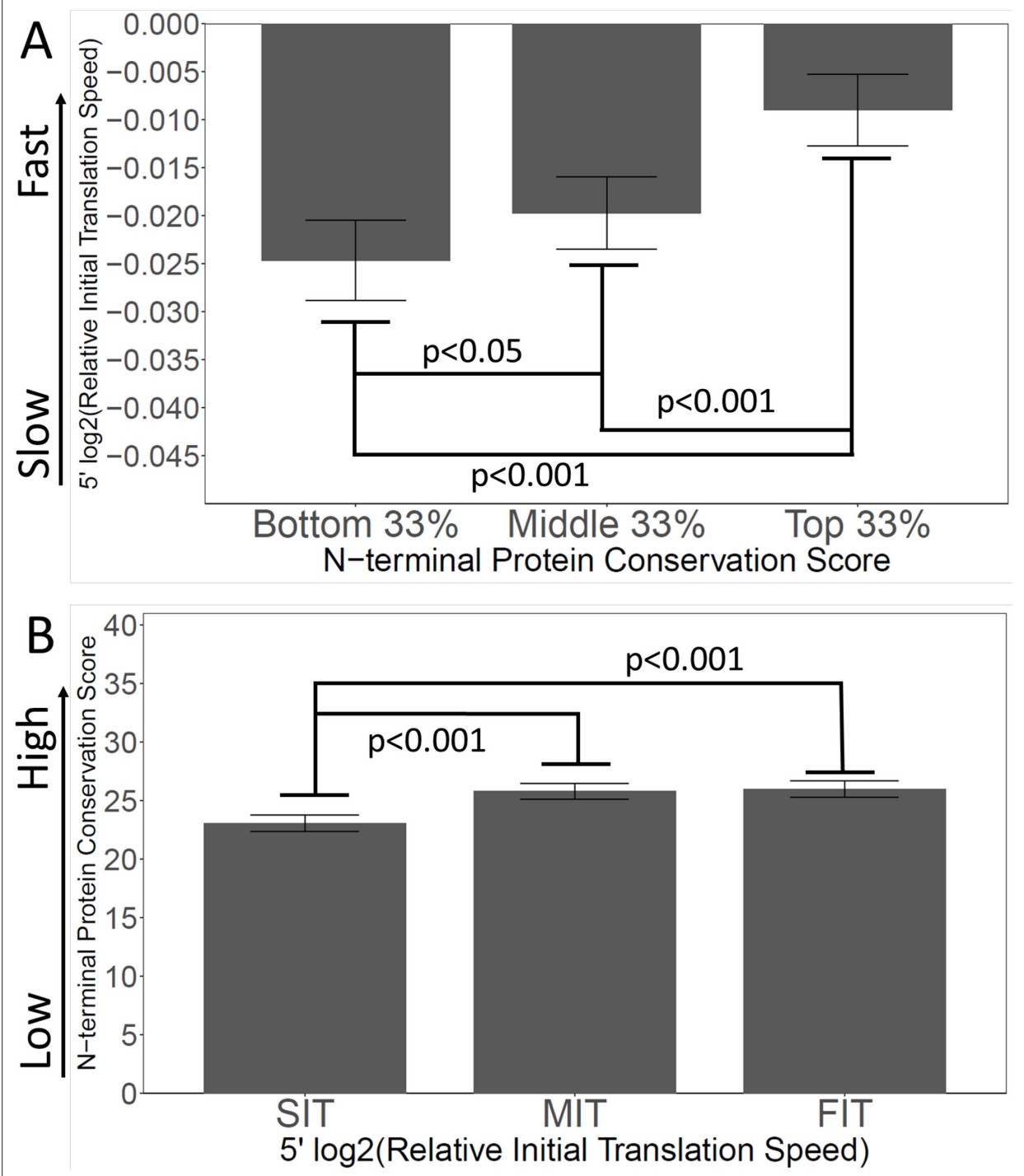

**Figure 6.** Slow initial translation is correlated with poor N-terminal conservation. (**A**) Proteins were grouped by their N-terminal conservation scores (top, middle, and bottom thirds), and then the relative initial translation rate was plotted for each group. More conserved N-termini have a faster initial translation. (**B**) Proteins were grouped by their initial translation rate (Slow, SIT; Medium, MIT, or Fast, FIT), and then the N-terminal conservation scores were plotted for each group. Genes with faster initial translation have more conserved N-termini. Relative Initial Translation Speed is the log2 of (average ribosome residence time, RRT of the first 40 amino acids divided by the average RRT of the rest of the same gene) (Methods and materials).

The online version of this article includes the following figure supplement(s) for figure 6:

**Figure supplement 1.** Slow 3′ translation is correlated with poor C-terminal conservation.

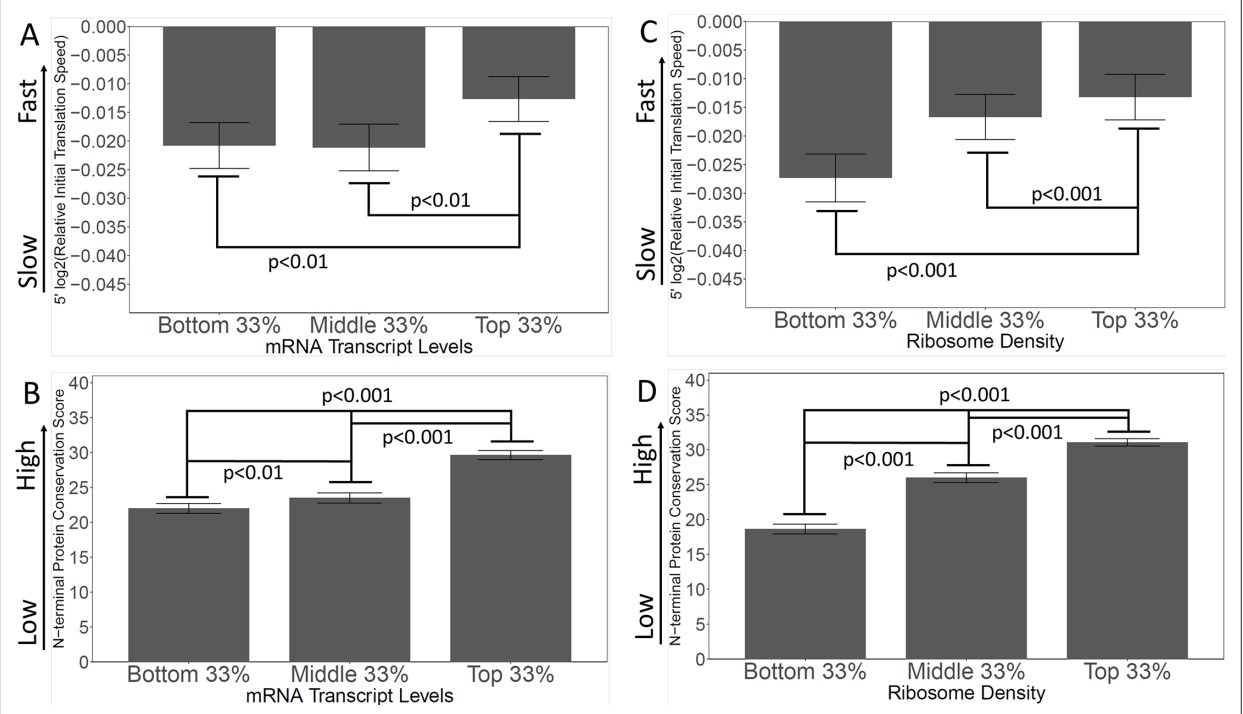

**Figure 7.** Genes with high levels of expression, and high ribosome densities, generally have rapidly-translated N-termini, and high N-terminal conservation scores. (**A** and **B**) Genes were grouped by expression level (bottom, middle, and top)(except that genes with fewer than 10 read-counts were omitted to reduce noise) (*Lipson et al., 2009*). In A, the initial translation rate is shown; in B, the conservation scores are shown. The correlation between speed and transcript abundance fails for the bottom third of genes; possibly these are genes expressed at high levels under other conditions (e.g. meiosis and sporulation). (**C** and **D**) Genes were grouped by ribosome density (*Arava et al., 2003*) as a measure of intensity of translation. In C, the initial translation rate is shown; in D, the conservation scores are shown. High ribosome density correlates with high initial translation speed and high conservation score.

We also looked at the correlation in the other direction (*Figure 6B*). We ranked genes by their relative initial translation speed, and divided the ranked list into thirds, then plotted N-terminal conservation scores. Again the effects are correlated: genes with the slowest initial translation have the lowest N-terminal conservation scores. Thus, overall, there is a strong correlation between N-terminal instability (i.e. newness in evolution, low conservation scores) and slow initial translation (i.e. the presence of slow/rare codons).

These correlations (i.e. between poor conservation and slow translation; and between slow translation and poor conservation) were also seen at the 3' ends of genes (*Figure 6—figure supplement 1*).

## The Ramp hypothesis is inconsistent with observations of ribosome density and gene expression

In the Tuller 'Ramp' hypothesis, in which the purpose of the slow translational ramp is to queue ribosomes and prevent collisions, genes with the highest ribosome occupancy would be in the most danger of ribosome collisions, and would, therefore, presumably have pronounced SITs. SITs might not be necessary on genes with low ribosome density, since there would not be much danger of collision in any case. To test this, we used information from *Arava et al., 2003*, which measured the density of ribosomes on all *S. cerevisiae* mRNAs (*Arava et al., 2003*; *Figure 7*). We ranked genes by ribosome density, then grouped them in thirds. Opposite to the expectation from the Tuller et al. 'Ramp' theory, the genes with the highest ribosome densities had the fastest initial translation, whereas the genes with the lowest ribosome densities had the slowest initial translation (*Figure 7C*). While these findings are opposite to the expectation of the 'Ramp' theory, they are consistent with the spandrel theory, because genes with high ribosome density would be subject to more intense selection against slow codons, thus leading to faster 5' ends. Furthermore, analysis of Conservation Scores on the same

genes showed that the genes with the lowest ribosome densities also had the lowest Conservation Scores (**Figure 7D**), as predicted by the Spandrel hypothesis.

Similarly, in the 'Ramp' hypothesis, ribosome collisions on highly-expressed genes would presumably have more serious consequences for the cell than collisions on poorly-expressed genes, and so highly-expressed genes ought to have the most pronounced SITs. To test this, we used transcriptomic information from **Lipson et al., 2009**, which measured the number of mRNA transcripts for *S. cerevisiae* genes (**Lipson et al., 2009**). Again, we ranked genes by expression, then grouped them by thirds. Exactly contrary to the 'Ramp' hypothesis, we found that genes with the highest expression had the fastest initial translation (**Figure 7A**). In this analysis, the middle and bottom genes are not significantly different from each other; possibly some of the poorly expressed genes are inducible genes that would be highly expressed under some other condition (e.g. the sporulation genes, the *GAL* genes). In addition, the most highly expressed genes had the highest conservation scores, consistent with the Spandrel hypothesis (**Figure 7B**).

## Experimentally, encoded slow initial translation does not increase gene expression; the opposite is true

Tuller et al. hypothesized that slow initial translation was adaptive, and improved the efficiency of translation and gene expression by minimizing ribosome collisions. However, in the Spandrel hypothesis, slow initial translation is not generated by selection and is not adaptive. It might not have any effect on gene expression, but, if anything, slow translation might reduce gene expression. While informatics is wonderful, it is always nice to do an experiment, and in this section, we present direct experimental results regarding the effect of encoded slow, medium, and fast initial translation on gene expression.

We used a gene expression reporter based on EKD1024 (**Brule et al., 2016**) (Methods and materials). In this construct, GFP is the reporter, but for accuracy it is normalized against a divergently transcribed red fluorescent protein (**Figure 8—figure supplement 1**). Thus, GFP expression is reported as a GFP/RFP ratio. Although the reporter is GFP, the N-terminal region of this particular protein is

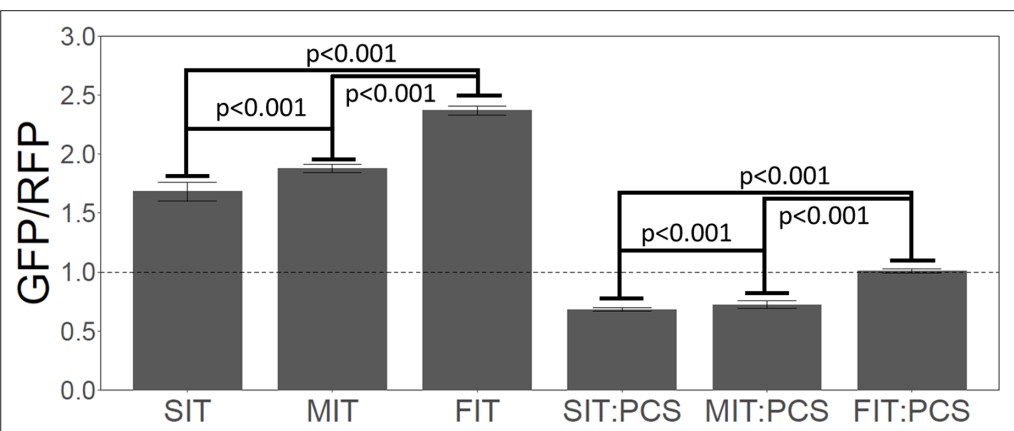

**Figure 8.** Slow initial translation inhibits gene expression. Left three bars. A synthetic GFP was constructed with a leader amino acid sequence that had little effect on GFP. The leader sequence was recoded to give slow (SIT), medium (MIT), or fast (FIT) translation speed over the first 41 amino acids, without changing the amino acid sequence—i.e., the SIT, MIT, and FIT had identical amino acid sequences, but different average ribosome residence times (RRTs). Each construct (SIT, MIT, FIT) was integrated in a single copy at the *ADE2* locus, and 25 independently-transformed strains were picked, and GFP fluorescence was measured for each, and the RFP-normalized mean was plotted. Numerical values were: SIT, 1.66; MIT, 1.80, FIT, 2.29. GFP was normalized to RFP expressed from the same reporter molecule, but RFP fluorescence hardly changed amongst the transformants, and non-normalized GFP would have given very similar results. Slower initial translation reduced gene expression. Right three bars. As above, a Putative ribosome collision site (PCS) (CGA-CGG) was inserted between the leader and the GFP. Again, slower initial translation reduced gene expression. Values were: SIT:PCS, 0.69, MIT:PCS, 0.74, FIT:PCS, 0.99.

The online version of this article includes the following figure supplement(s) for figure 8:

**Figure supplement 1.** Structure of the GFP reporters.

derived from yeast *HIS3*, not GFP, and likely has little if any effect on the fluorescence of the GFP fused downstream (*Dean and Grayhack, 2012*; *Gamble et al., 2016*; *Pédelacq et al., 2006*). We used synonymous slow, medium, or fast codons to recode some of the codons in the first 41 amino acids of this GFP reporter to generate three reporters with slow, medium, or fast translation over the first 41 amino acids. We emphasize that the amino acid sequences of the three constructs were identical. The slow, medium, and fast average RRT values over the first 41 codons were 1.20, 1.04, and 0.93, respectively. That is, this SIT is slower than most natural SITs, and this FIT is faster than most natural FITs, but the difference is moderate.

As shown in *Figure 8* (left three bars), the SIT did not improve expression of GFP, contrary to Tuller et al. In fact, the GFP with the SIT was expressed at only 71% of the level of the GFP with the FIT. It was surprising to us that the difference was this large—again, recoding was limited to codons within the first 41, and the protein sequences were identical.

Another possibility is that a SIT can protect against ribosome collisions when there is a site downstream that induces ribosome collisions. Sites thought to induce ribosome collisions include rare Arg-Arg codon pairs (*Dao Duc and Song, 2018*; *Tesina et al., 2020*). We, therefore, introduced the codon pair CGA-CGG (replacing Asn-Asp, AAT-GAT) downstream of the first 41 amino acids, but still upstream of important GFP residues. Indeed, this single CGA-CGG codon pair, potentially inciting ribosome collisions, caused a large reduction--about 50%--in the expression of GFP (*Figure 8*, right three bars). The reduction was about the same in the SIT, MIT, and FIT constructs. In this case, with putative collision sites, the GFP with the SIT was expressed at only 67% of the level of the equivalent GFP with the FIT. That is, this SIT (a fairly extreme SIT) did not at all protect against the putative ribosome collisions—if anything, it made things slightly worse. This result suggests there is no benefit to 'queuing' ribosomes, if queuing even occurs. Instead, the fastest-translating gene once again gave the highest expression, and the highest relative expression, despite the collision site.

## Discussion

Tuller et al. found that the 5' ends of genes are translated slowly because of the codons used at 5' ends, and posited that this was a selective advantage because it somehow increased the efficiency of translation. However, this theory predicts positive correlations between slow initial translation and high gene expression, and slow initial translation and high overall (that is, on the whole gene) ribosome density. In fact, by informatic analysis of existing data, we find the correlations are opposite to those predicted by the 'Ramp' model. Most importantly, an experiment in which codon usage at 5' ends was changed shows that faster 5' codons cause higher gene expression, exactly opposite to the prediction of the 'Ramp' hypothesis. We believe no ramp is needed.

Tuller et al. showed that a region of slow translation is encoded, using slowly-translated codons, and it is specifically this idea of encoded slow translation that we are addressing. This encoded slow translation is a small effect—translation is slowed by 1% to 3%. However, in addition to 'encoded' slow translation, there is evidence for slow translation at 5' ends by other, unknown mechanisms, with apparently much larger amplitudes, perhaps greater than 50%. Ribosome profiling experiments show an increased density of ribosome footprints near the 5' end, independent of encoding (*Weinberg et al., 2016*), which could be due to slow translation. It is now known that the very high 5' density of footprints in early ribosome profiling studies was due to the use of cycloheximide as a first step to stop translation. Addition of cycloheximide to growing cells allowed ribosomes to initiate at Start codons, but did not allow elongation, hence there was a pile-up of ribosomes near the 5' end. More recently, flash-freezing, rather than cycloheximide, has been used as the first step in stopping translation. However, even in these studies, there is about a 50% increase in ribosome density near 5' ends (*Weinberg et al., 2016*). And yet, even in these flash-freezing protocols, cycloheximide is still used at a later step to prevent elongation when extracts are thawed, and this cycloheximide usage could again result in an artifactual increase in ribosome density at 5' ends. Alternatively, the increased density of ribosomes at 5' ends could mean that some proportion of ribosomes fall off the mRNA as they progress (*Weinberg et al., 2016*). Consistent with the latter idea, ribosome profiling studies show a general trend towards lower ribosome densities at more 3' positions in translating mRNAs (Weinberg et al., their Figure S7), and studies using other experimental approaches have shown a general decrease in ribosome number or density as one progresses along a gene (*Bonderoff and Lloyd, 2010*; *Verma et al., 2019*). A different idea was proposed by *Shah et al., 2013* in a theory paper, which suggested

this apparent slow translation could be an informatic artifact caused by rapid translational initiation (and, therefore, high ribosome density) on short genes. But none of these ideas addresses the fact found by Tuller that the 5′ ends of genes are enriched in rare, slow codons.

We considered that an increased density of ribosomes at the 5′ end could be because some genes have additional ATG Start codons, sometimes upstream and sometimes downstream of the annotated Start, and translation of short open reading frames from these additional Start codons could contribute to ribosome density at the 5′ end. Using the program 'Frameshift Detector' (*Yurovsky et al., 2022*) and ribosome profiling data, we quantitated the fraction of out-of-frame ribosomes both globally, and within the first 150 nucleotides of genes. We found the global proportion of out-of-frame ribosomes is about 13%, and the proportion of out-of-frame ribosomes in the first 150 nucleotides is about 14.5%. Although this increased 5′ out-of-frame ribosome presence of about 1.5% is highly significant ($p \sim 10^{-28}$), it is not nearly big enough to explain the observed 5′ increase in ribosome density (*Weinberg et al., 2016*, their Figure 1C).

In any case, by direct experiment, we find that encoding a slower 5′ end using slow synonymous codons reduces gene expression. In particular, even when a ribosome collision site was placed downstream, the effect of the collision site was not at all ameliorated by encoded slow translation upstream of the collision site. This seems strong evidence against the idea that slow initial translation is a defense against collisions. The basis of the idea that slow initial translation could possibly be a defense against collisions is not clear to us. Regions of slow translation would not affect the gaps between ribosomes, if measured as times, and especially not if measured at a constant finish line, such as a putative collision site.

An issue in the GFP reporter experiment is that the mRNA sequences are necessarily different, and so there are different mRNA structures. We achieved fast and slow sequences by recoding with synonymous codons, so amino acids are identical, so there are no issues of, e.g., co-translational protein folding, or amino acid interaction with the ribosome exit tunnel. But of course, the RNA structures are at least slightly different, and RNA structures at the 5′ end are known to affect translation initiation (*Weinberg et al., 2016*; *Burkhardt et al., 2017*; *Cuperus et al., 2017*; *Gu et al., 2010*; *Hall et al., 1982*; *Kudla et al., 2009*; *Nackley et al., 2006*). Generally, more open mRNA structures are more favorable both for translation initiation and for translation speed. To fully disentangle these effects is difficult. But whether the increased gene expression we see for the fast encoding is being generated mainly by fast translation, or by efficient initiation, in neither case is there an argument that slow translation is efficient, or that it protects against collisions.

The observation that 5′ ends have low conservation, likely because of instability in evolution, provides a completely different explanation for the enrichment of slow codons at 5′ ends. In this 'Spandrel' hypothesis, N-termini frequently change in evolution, gathering new 5′ sequences de novo. These would contain all codons at similar frequencies—i.e., 'rare' codons would not be especially rare. Although rare codons would eventually be removed by selection, the fact that N-termini are relatively young means that this process might not be complete for all genes, and so some rare, slow codons still remain. These explain the initial region of encoded slow translation. This hypothesis is highly consistent with the observed correlations between slow initial translation and low gene expression; and slow initial translation and low ribosome density, and with the results of gene expression experiments. It is also consistent with the region of slightly slow translation we observe at 3′ ends.

We have looked at the conservation of N-termini only in *S. cerevisiae*. However, Tuller et al. found that there is a region of encoded slow initial translation in genes of a wide variety of eukaryotes. We speculate that in these other cases, too, slow initial translation is a spandrel partly due to depletion of fast Start and alternative Start codons, and partly deriving from the turnover of 5′ ends. This in turn has implications for protein structure and evolution; for the interpretation of evolutionary sequence clocks; and for the rates of selection against rare codons. We note that *Bricout et al., 2023* have also recently found that N- and C-termini of proteins evolve faster than the middles.

It was surprising to us that recoding just the first 41 codons of the GFP fusion protein from slow to fast increased the level of GFP expression by so much—about 30%. These 41 codons were originally derived from the yeast *HIS3* gene, and this increase in expression is roughly the proportional increase expected based on fully recoding the *HIS3* gene to preferred codons (*Presnyak et al., 2015*). Because this recoding from slow to fast tends to replace G/C-rich codons with A/T-rich codons, recoding from slow to fast may decrease the stability of RNA structures near the 5′ end, and increase the accessibility

of the cap. Decreased stability of mRNA structures could be responsible for the increase in gene expression, consistent with studies in both yeast (*Weinberg et al., 2016*; *Cuperus et al., 2017*) and *E. coli* (*Kudla et al., 2009*).

Finally, again, in 'The Spandrels of San Marco...,' (*Gould and Lewontin, 1979*) warned that not all biological phenomena are adaptive, and it is a mistake to assume that any particular characteristic of an organism must necessarily have been generated by natural selection. We believe the encoded slow initial translation of eukaryotic genes may be an example of this.

## Materials and methods
### Statistical calculations of relative initial rate of translation
Bioinformatics were performed on protein-coding open reading frames (ORFs) of *Saccharomyces cerevisiae* downloaded from the *Saccharomyces* Genome Database (SGD) website, as last modified on April 22, 2021. Protein-coding ORFs annotated as dubious or pseudogenes were not included in analyses. All statistics were performed using The R Project for Statistical Computing. Translation speed was measured using the ribosome residence time (RRT) which is a metric of the occupancy of ribosomes on each sense codon within the A-site (*Gardin et al., 2014*). The RRT values we used are shown in *Supplementary file 1*; these are modified from the original RRT results of Gardin et al. by inclusion of the ribosome profiling data of *Jan et al., 2014*.

### Relative initial translation speed
(*Tuller et al., 2010*) focused on a 'ramp' of translation speed, where the first part of the gene has slow translation relative to the rest of the gene. The ramp thus refers to a rate. To quantitate this slow relative ramp for each gene, we calculated the average RRT for an initial window of the gene (e.g. 40 amino acids, see below), then divided by the average RRT of the rest of the gene. Thus, genes with a 'slow ramp' have a ratio of less than 1. We then took log2 of this ratio; genes with a slow ramp yield a negative number, and the more negative the number, the steeper the ramp.

For each gene, the relative initial translation speed (RIT) (explained above) was calculated across windows of the first 30, 40, 50... and 100 codons, with all windows being statistically significant for slow translation. For these RIT calculations, the first (start) codon was omitted since all protein-coding genes in this dataset except Q0075 start with ATG, and ATG is one of the fastest codons, which would skew the RIT. Similarly, the last (stop) codon was omitted.

$$\text{RIT} = \log2\left(\frac{\text{meanRRT}\left(\text{codons}\,2:\text{window}\right)}{\text{meanRRT}\left(\left(\text{codons}\left(\text{window}+1\right)\right):\left(\text{last codon}-1\right)\right)}\right)$$

Windows of the first 30, 40, 50... and 100 codons each had statistically significant depletion in translation speed compared to the body of genes. We chose to focus on the first 40 codons. All genes shorter than 303 nucleotides (translated into 100 amino acids) were omitted from all analyses. For all RIT analyses, 328 out of 6022 ORFs were omitted leaving a dataset of 5694 ORFs.

### Data
mRNA transcript readings, which we used as a proxy for gene expression, was acquired from *Lipson et al., 2009*. Genes with a read count of less than 10 were omitted due to concerns about noise. Ribosome density measurements were acquired from *Arava et al., 2003*. These values were calculated as the number of ribosomes, detected on an mRNA, divided by the nucleotide length of the gene (including the stop codon).

### Protein BLAST setup and diagnostics
*S. cerevisiae* proteins were downloaded from the SGD (last modified on April 22, 2021). Proteins derived from ORFs annotated as dubious or pseudogene were omitted from analyses. The *Saccharomycotina* (Taxonomy ID: 147537) protein sequences were downloaded from NCBI using the links here.

We downloaded and compiled the source databases from DDBJ, EMBL, Genbank, RefSeq, PIR, and UniProtKB. Duplicate sequences were deleted. To perform local BLAST, we downloaded the NCBI BLAST software (version 2.13.0+) and used RStudio as a wrapper to operate the software; all

default BLAST parameters were selected, except that the number of alignments was changed to the maximum value of 1000000000. Local protein BLAST of every *S. cerevisiae* protein (5694 proteins, see above) was performed against all genomes of the subphylum *Saccharomycotina*, but omitting all species in the genus *Saccharomyces* (net, 822 genomes). We eliminated submissions of duplicate species by limiting our database to the highest bit-scores from sequence hits derived from each unique species. We were only interested in sequences that had high homology with queried *S. cerevisiae* proteins, so all hits with bit-scores lower than 50 were omitted.

We wanted to compare conservation at the beginning of proteins with conservation at the middle and end of those same proteins. For this purpose, we split each *S. cerevisiae* protein into two halves (start to middle; middle to end), then blasted each half against all genomes in the subphylum *Saccharomycotina* (omitting *Saccharomyces*). We then calculated a 'conservation score' (see below) for the first 40 amino acids of the protein, and, identically, for the first 40 amino acids of the second half of the protein. (We describe the first 40 amino acids of the second half of the protein as the 'middle,' but in fact, the region is displaced 20 amino acids C-terminal from the exact middle.) In a parallel way, a conservation score is calculated for the last 40 amino acids of each protein. For example, the length of Swi5 is 709 amino acids, and therefore BLASTs of the first half spanned from 1:354, and BLASTs for the second half spanned from 355:709. Conservation scores were calculated for residues 1:40 (beginning) (from the BLASTs of the first half of the protein), and 355:394 (middle) and 670:709 (end) (from BLASTs of the second half of the protein). In total, BLAST of the first half of all queried *S. cerevisiae* proteins yielded a total of 477,749 high homology (minimum of bit-score of 50) sequence matches across 816 unique *Saccharomycotina* species, whereas BLAST of the second half of each protein yielded a total of 487,022 high homology matches across 812 unique *Saccharomycotina* species.

With respect to the above procedure, we note that we are relying on the BLAST algorithm to find regions of homology. Homology would be somewhat more easily found in the middle of sequences than at the ends because of seeding issues. It is for this reason that we divided proteins in half, and used a BLAST with the second half of the protein to find homologies with the first 40 amino acids of the second half. That is, in this procedure, for the middle homologies, the algorithm is being asked to find homologies at the end of a sequence, exactly as is the case for the first 40 and last 40 amino acids. We also used the alternative approach of finding homologies in the last 40 amino acids of the first half of the protein, with essentially identical results (*Figure 4—figure supplement 1*).

## Calculations of protein conservation scores and ratios

The general idea of the 'Conservation Score' is that it represents the lengths of the regions of BLAST homology between the *S. cerevisiae* query and the *Saccharomycotina* subjects in the beginning, middle, and end 40-aminoacid windows. Each *S. cerevisiae* query sequence was separated into two equal halves, and then BLASTs were done on both halves against all of *Saccharomycotina* (omitting all submissions from the *Saccharomyces* genus). For all subject proteins with high homology over any region (i.e. a bit-score greater than 50), one finds the pair-wise regions of homology with a BLAST 'Alignment Score' of 200 or more (red colored regions in the BLAST website 'Graphic Summary'). The length of the high-homology region in the window of interest (but not the actual number of amino acid sequence matches within that region) contributes to the Conservation Score. That is, amino acids have to be within a region of homology found by BLAST in order to contribute to the score; even though all proteins begin with 'M,' these only contribute to the conservation score if they are within a region of BLAST homology. The Conservation Score is the sum of (the length of the homology in the window, multiplied by the proportion of qualified subject proteins with that length of homology). Example Conservation Scores are shown in *Supplementary file 3*, and an example Conservation Score is calculated in *Supplementary file 4*. We only considered conservation scores from proteins that had homology with at least 40 unique species in *Saccharomycotina*. We also omitted proteins that were shorter than 100 amino acids. As shown in *Supplementary file 3*, conservation scores ranged from 0, meaning no BLAST homology region within the window of 40 amino acids for any qualifying homolog in *Saccharomycotina*, up to a maximum of 40, meaning that all qualifying homologs in *Saccharomycotina* had matches starting at the first amino acid. In total, protein conservation analyses used 3964 *S. cerevisiae* proteins with high homology hits for BLAST done on the first and second half of proteins (*Figure 4*).

## Design of the fluorescent reporter gene constructs

We created a reporter gene based on reporter plasmid EKD1024 (*Brule et al., 2016*). Briefly, a bidirectional galactose promoter simultaneously induces the expression of GFP and RFP in the presence of galactose, and we integrated the reporter into the yeast genome at the *ADE2* locus. We recoded GFP to give the first 41 codons of GFP a slow initial translation speed (SIT), medium initial translation speed (MIT), or fast initial translation speed (FIT), while maintaining the same amino acid sequence (*Figure 8—figure supplement 1*, sequences in *Supplementary file 5*). We also designed three more constructs (*Figure 8—figure supplement 1*, sequences in *Supplementary file 5*) with a SIT, MIT, or FIT upstream of one of the slowest and rarest codon pairs, CGA-CGG (replacing AAT-GAT, Asn-Asp), which is known to greatly attenuate gene expression in living yeast (*Gamble et al., 2016*). Other rare codon pairs (CGA-CGA and CGA-CCG) have been shown to promote ribosome stalling (*Tesina et al., 2020*) so in *Figure 8*, right, CGA-CGG operates as a putative ribosome collision site (PCS). Instead of a PCS, the constructs in *Figure 8*, left, had AAT-GAT (Asn Asp), a frequent codon pair with above average translation speed. The Relative Initial Translation Speed scores of the constructs were: SIT was 0.208; MIT was –0.0004; FIT was –0.166; SIT + PCS was 0.198; MIT + PCS was –0.0109; and FIT + PCS was –0.177. (Note that the RIT scores of the constructs with the PCS change because the PCS makes the translation speed of the body of the gene slower; that is, the change is due to a change in the denominator.)

## Yeast strains

The constructs were transformed into BY4741 (*MATa his3Δ1 leu2Δ0 met15Δ0 ura3Δ0*). The reporter expresses *MET15* allowing selection. Transformants were selected for Met+ on HULA plates (0.075 g/L Histidine; 0.075 g/L Uracil; 0.25 g/L Leucine; 0.075 g/L Adenine; 20 g/L D-Glucose; 5 g/L Ammonium Sulfate; 1.7 g/L Yeast Nitrogen Base). The reporter gene integrates into the *ADE2* locus, and thus successful transformants are *ade2*-delete which becomes red when grown on YPD plates; this was used as a secondary biological marker to confirm successful transformants. About 30 Met+, Ade-, red transformants were chosen for each recoded GFP construct. These transformants were pre-screened using a flow cytometer for absolute levels of galactose-induced green and red fluorescence; out of all the individual colonies initially chosen (~180), about 5 were rejected because their absolute levels of both GFP and RFP fluorescence were about twice as high as for other strains. We believe these rejected transformants contained two copies of the reporter construct. For each construct, 25 Met+, Ade-, red transformants were chosen for analysis. Strains are available upon request to BF, and sequences of SIT, MIT, and FIT constructs are available in *Supplementary file 5*.

## Flow cytometry analysis

The strains were inoculated in liquid HULA media, with 2% galactose, until mid-log phase (around 10–14 hr) containing around $3 \times 10^7$ cells/mL. The strains were sonicated to separate cells and the strains were stored on ice (typically around an hour) until data collection. An LSR Fortessa Flow Cytometer was used to measure GFP levels (All Events FITC-A Mean) and RFP levels (All Events PE-Texas Red-A Mean) across 75,000 events. As a control, a strain lacking a fluorescent reporter was used. For all samples, GFP levels were normalized by RFP levels. All samples (25 per experiment) were included in the analysis with no exclusions.

## Statistical tests

None of our analyses made assumptions regarding the normality of the data. As such, we only performed nonparametric statistics. Wilcoxon signed-rank tests were done for relevant pairwise analyses; when necessary, p-values were corrected for multiple comparisons using the Holm–Bonferroni method. Spearman correlations were used. We used the Kolmogorov–Smirnov goodness of fit test to confirm that the three distributions were significantly different ($p<0.001$) for *Figure 4*.

## Acknowledgements

We thank Steve Ketchum and Sangeet Honey for discussions that helped form the central idea of this manuscript.

## Additional information

### Funding

| Funder | Grant reference number | Author |
|---|---|---|
| National Institutes of Health | RO1 GM127542 | Bruce Futcher |
| National Institutes of Health | RO1 GM 132238 | Bruce Futcher |

The funders had no role in study design, data collection and interpretation, or the decision to submit the work for publication.

### Author contributions

Richard Sejour, Conceptualization, Data curation, Software, Formal analysis, Validation, Investigation, Visualization, Methodology, Writing - original draft, Project administration, Writing - review and editing; Janet Leatherwood, Conceptualization, Formal analysis, Supervision, Validation, Investigation, Visualization, Methodology, Writing - review and editing; Alisa Yurovsky, Software, Formal analysis, Investigation, Methodology, Writing - review and editing; Bruce Futcher, Conceptualization, Formal analysis, Supervision, Funding acquisition, Validation, Investigation, Visualization, Methodology, Writing - original draft, Project administration, Writing - review and editing

### Author ORCIDs

Bruce Futcher  http://orcid.org/0000-0002-1012-9022

Reviewer #1 (Public Review): https://doi.org/10.7554/eLife.89656.3.sa1
Reviewer #2 (Public Review): https://doi.org/10.7554/eLife.89656.3.sa2
Author response https://doi.org/10.7554/eLife.89656.3.sa3

## Additional files

### Supplementary files

• Supplementary file 1. Ribosome residence time (RRT) values. See attached Excel spreadsheet.

• Supplementary file 2. RRT statistics of various 5' ends. See attached Excel Spreadsheet.

• Supplementary file 3. Example conservation scores. Scores were calculated as described in Materials and methods, but in this example, only for a subset of *Saccharomycotina*. 'Total Hits' is the number of different proteins from the sub-phylum *Saccharomycotina* subset giving a BLAST bit-score of at least 50. 'Hits in the first 40 amino acids' is the number of proteins (out of the proteins in the 'Total Hits' columns) that had a BLAST alignment with an alignment score >200 matching any part of the first 40 amino acids of the query sequence (i.e. of *PCA1*, *NSR1*, etc.). 'Query Start' is the range of amino acid positions in the Query protein where the BLAST alignments started. For instance, for *BUD5*, the 125 *Saccharomycotina* homologs had BLAST alignments that started at positions between amino acid 211 and amino acid 420 on *S. cerevisiae BUD5*; none had an alignment starting within the first 40 amino acids. For *SNX41*, 65 of the 121 hits had an alignment beginning within the first 40 amino acids of *S. cerevisiae SNX41*. For *RPL12B*, all 121 of the *Saccharomycotina* homologs had BLAST alignments starting at amino acid 1 of *S. cerevisiae RPL12B*. The 'Conservation Score' is the score calculated as described in Materials and methods. Note that the number of hits varies in part because the genomes of the *Saccharomycotina* species were not all fully sequenced. Thus, *BNA2* likely has fifteen fewer hits than *TRP3* because the *BNA2* locus was not sequenced in some species. However, the number of hits does not affect the conservation score, as long as the number meets the qualifying minimum.

• Supplementary file 4. Calculation of a conservation score. 'Query start position in BLAST alignment' is the amino acid residue of the *S. cerevisiae* query protein where a BLAST alignment (alignment score >200) begins with a protein of *Saccharomycotina*. 'Proportion of hits with this Q-Start Position' is the proportion of qualifying *Saccharomycotina* hits (i.e. bit-score >50) that have their alignment begin at this position. 'Weight' is multiplied by 'Proportion,' and the sum is the conservation score.

- Supplementary file 5. Sequences of Ramp genes in *Figure 8*.
- MDAR checklist
- Source code 1. We provide the custom R code written for this project as a text file, Source Code File 1.

### Data availability

All data generated or analysed during this study are included in the manuscript and supporting files.

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
