## [Editor Report · eLife assessment]

This is an **important** contribution to the origins and translational consequences of the relatively low rate of translation elongation in the first ∼30-50 codons of genes in most organisms. The authors provide **convincing** evidence that the prevalence of rare codons in the first ~40 codons in yeast is due to the relatively recent evolution of these coding sequences, or of lower purifying selection operating on them, and that a preponderance of codons encoded by rare tRNAs near the N-terminus is not associated with higher translational efficiency in the manner proposed by the "translational ramp" hypothesis. The work is **incomplete** in that the results of reporter assays may have been confounded by alterations of mRNA sequence or structure that could have influenced their translation or mRNA stability; that the work cannot fully account for a greater enrichment of slowly translated codons in N-terminal vs. C-terminal regions; and that the work does not resolve whether translation elongation through N-terminal coding is truly slow.

---

## [Referee Report · Reviewer #1 (Public Review)]

The manuscript by Sejour et al. is testing "translational ramp" model described previously by Tuller et al. in *S. cerevisiae*. Authors are using bioinformatics and reporter based experimental approaches to test whether "rare codons" in the first 40 codons of the gene coding sequences increase translation efficiency and regulate abundance of translation products in yeast cells. Authors conclude that "translation ramp" model does not have support using a new set of reporters and bioinformatics analyses. The strength of bioinformatic evidence and experimental analyses (even very limited) of the rare codons insertion in the reporter make a compelling case for the authors claims. However the major weakness of the manuscript is that authors do not take into account other models that previously disputed "rare or slow codon" model of Tuller et al. and overstate their own results that are rather limited. This maintains to be the weak part of the manuscript even in the revised form.

The studies that authors do not mention argue with "translation ramp" model and show more thorough analyses of translation initiation to elongation transition as well as early elongation "slow down" in ribosome profiling data. Moreover several studies have used bioinformatical analyses to point out the evolution of N-terminal sequences in multiple model organisms including yeast, focusing on either upstream ORFs (uORFs) or already annotated ORFs. The authors did not mention multiple of these studies in their revised manuscript and did not comment on their own results in the context of these previous studies. As such the authors approach to data presentation, writing and data discussion makes the manuscript rather biased, focused on criticizing Tuller et al. study and short on discussing multiple other possible reasons for slow translation elongation at the beginning of the protein synthesis. This all together makes the manuscript at the end very limited.

---

## [Referee Report · Reviewer #2 (Public Review)]

Tuller et al. first made the curious observation, that the first ∼30-50 codons in most organisms are encoded by scarce tRNAs and appear to be translated slower than the rest of the coding sequences (CDS). They speculated that this has evolved to pace ribosomes on CDS and prevent ribosome collisions during elongation - the "Ramp" hypothesis. Various aspects of this hypothesis, both factual and in terms of interpreting the results, have been challenged ever since. Sejour et al. present compelling results confirming the slower translation of the first ~40 codons in *S. cerevisiae* but providing an alternative explanation for this phenomenon. Specifically, they show that the higher amino acid sequence divergence of N-terminal ends of proteins and accompanying lower purifying selection (perhaps the result of de novo evolution) is sufficient to explain the prevalence of rare slow codons in these regions. These results are an important contribution in understanding how aspects of the evolution of protein coding regions can affect translation efficiency on these sequences and directly challenge the "Ramp" hypothesis proposed by Tuller et al.

I believe the data is presented clearly and the results generally justify the conclusions.

---

## [Author Response]

The following is the authors’ response to the original reviews.

Response to Reviewers:

We thank the reviewers for their comments, and their evident close reading of the manuscript.Generally, we agree with the reviewers on the strengths and weaknesses of our manuscript. Ourrevised manuscript has a more extensive discussion of alternative explanations for initial highribosome density as seen by ribosome profiling, and which more specifically points out thelimitations of our work.

As a preface to specific responses to the reviewers, we will say that we could divide observationsof slow initial translation into two categories, which we will call “encoded slow codons”, and“increased ribosome density”. With respect to the first category, Tuller et al. documented initial“encoded slow codons”, that is, there is a statistical excess of rare, slowly-translated codons atthe 5’ ends of genes. Although the size of this effect is small, statistical significance is extremelyhigh, and the existence of this enrichment is not in any doubt. At first sight, this appears to be astrong indication of a preference for slow initial translation. In our opinion, our maincontribution is to show that there is an alternative explanation for this initial enrichment of rare,slow codons—that they are a spandrel, a consequence of sequence plasticity at the 5’ (and 3’)ends of genes. The reviewers seem to generally agree with this, and we are not aware that anyother work has provided an explanation for the 5’ enrichment of rare codons.

The second category of observations pertaining to slow initial translation is “increased ribosomedensity”. Early ribosome profiling studies used cycloheximide to arrest cell growth, and thesestudies showed a higher density of ribosomes near the 5’ end of genes than elsewhere. This highinitial ribosome density helped motivate the paper of Tuller et al., though their finding of“encoded slow codons” could explain only a very small part of the increased ribosome density.More modern ribosome profiling studies do not use cycloheximide as the first step in arrestingtranslation, and in these studies, the density of ribosomes near the 5’ end of genes is greatlyreduced. And yet, there remains, even in the absence of cycloheximide at the first step, asignificantly increased density of ribosomes near the 5’ end (e.g., Weinberg et al., 2016).(However, most or all of these studies do use cycloheximide at a later step in the protocol, andthe possibility of a cycloheximide artefact is difficult to exclude.) Some of the reviewer’sconcerns are that we do not explain the increased 5’ ribosome density seen by ribosomeprofiling. We agree; but we feel it is not the main point of our manuscript. In revision, we moreextensively discuss other work on increased ribosome density, and more explicitly point out thelimitations of our manuscript in this regard. We also note, though, that increased ribosomedensity is not a direct measure of translation speed—it can have other causes.

Specific Responses.

Reviewer 1 was concerned that we did not more fully discuss other work on possible reasons forslow initial translation. We discuss such work more extensively in our revision. However, as faras we know, none of this work proposes a reason for the 5’ enrichment of rare, slow codons, andthis is the main point of our paper. Furthermore, it is not completely clear that there is any slowinitial translation. The increase in ribosome density seen in flash-freeze ribosome profiling couldbe an artefact of the use of cycloheximide at the thaw step of the protocols; or it could be a real measure of high ribosome density that occurs for some other reason than slow translation (e.g.,ribosomes might have low processivity at the 5’ end).

Reviewer 1 was also concerned about confounding effects in our reporter gene analysis of theeffects of different codons on efficiency of translation. We have two comments. First, it isimportant to remember that although we changed codons in our reporters, we did not change anyamino acids. We changed codons only to synonymous codons. Thus at least one of thereviewer’s possible confounding effects—interactions of the nascent peptide chain with the exitchannel of the ribosome—does not apply. However, of course, the mRNA nucleotide sequenceis altered, and this would cause a change in mRNA structure or abundance, which could matter.We agree this is a limitation to our approach. However, to fully address it, we feel it would benecessary to examine a really large number of quite different sequences, which is beyond thescope of this work. Furthermore, mRNAs with low secondary structure at the 5’ end probablyhave relatively high rates of initiation, and also relatively high rates of elongation, and it mightbe quite difficult to disentangle these. But in neither case is there an argument that slow initialtranslation is efficient. Accurate measurement of mRNA levels would be helpful, but would notdisentangle rates of initiation from rates of elongation as causes of changes in expression.

Reviewer 2 was concerned that the conservation scores for the 5’ 40 amino acids, and the 3’ 40amino acids were similar, but slow translation was only statistically significant for the 5’ 40amino acids. As we say in the manuscript, we are also puzzled by this. We note that 3’translation is statistically slow, if one looks over the last 100 amino acids. Our best effort at anexplanation is a sort of reverse-Tuller explanation: that in the last 40 amino acids, the new slowcodons created by genome plasticity are fairly quickly removed by purifying selection, but thatin the first 40 amino acids, for genes that need to be expressed at low levels, purifying selectionagainst slow codons is reduced, because poor translation is actually advantageous for thesegenes. To expand on this a bit, we feel that the 5000 or so proteins of the proteome have to beexpressed in the correct stoichiometric ratios, and that poor translation can be a useful tool tohelp achieve this. In this explanation, slow translation at the 5’ end is bad for translation (inagreement with our reporter experiments), but can be good for the organism, when it occurs infront of a gene that needs to be expressed poorly. Whereas, in Tuller, slow translation at the 5’end is good for translation.

Reviewer 2 wondered whether the N-terminal fusion peptide affects GFP fluorescence in ourreporter. This specific reporter, with this N-terminus, has been characterized by Dean andGrayhack (2012), and by Gamble et al. (2016), and the idea that a super-folder GFP reporter isnot greatly affected by N-terminal fusions is based on the work of Pedelacq (2006). None ofthese papers show whether this N-terminal fusion might have some effect, but together, theyprovide good reason to think that any effect would be small. These citations have been added.